# Skin Microbiome in Prurigo Nodularis

**DOI:** 10.3390/ijms24087675

**Published:** 2023-04-21

**Authors:** Klaudia Tutka, Magdalena Żychowska, Anna Żaczek, Karolina Maternia-Dudzik, Jakub Pawełczyk, Dominik Strapagiel, Jakub Lach, Adam Reich

**Affiliations:** 1Department of Dermatology, Institute of Medical Sciences, Medical College of Rzeszow University, 35-055 Rzeszów, Poland; 2Department of Microbiology, Institute of Medical Sciences, Medical College of Rzeszow University, 35-055 Rzeszów, Poland; 3Laboratory of Genetics and Physiology of Mycobacterium, Institute of Medical Biology, Polish Academy of Sciences, 90-235 Łódź, Poland; 4Biobank Laboratory, Department of Oncobiology and Epigenetics, Faculty of Biology and Environmental Protection, University of Lodz, 90-235 Łódź, Poland

**Keywords:** microbiome, microbiota, prurigo nodularis, bacteria, 16S rRNA, DNA sequencing

## Abstract

Prurigo nodularis (PN) is a chronic condition characterized by the presence of nodular lesions accompanied by intense pruritus. The disease has been linked to several infectious factors, but data on the direct presence of microorganisms in the lesions of PN are scarce. The aim of this study was to evaluate the diversity and composition of the bacterial microbiome in PN lesions by targeting the region V3-V4 of 16S rRNA. Skin swabs were obtained from active nodules in 24 patients with PN, inflammatory patches of 14 patients with atopic dermatitis (AD) and corresponding skin areas of 9 healthy volunteers (HV). After DNA extraction, the V3-V4 region of the bacterial 16S rRNA gene was amplified. Sequencing was performed using the Illumina platform on the MiSeq instrument. Operational taxonomic units (OTU) were identified. The identification of taxa was carried out using the Silva v.138 database. There was no statistically significant difference in the alpha-diversity (intra-sample diversity) between the PN, AD and HV groups. The beta-diversity (inter-sample diversity) showed statistically significant differences between the three groups on a global level and in paired analyses. *Staphylococcus* was significantly more abundant in samples from PN and AD patients than in controls. The difference was maintained across all taxonomic levels. The PN microbiome is highly similar to that of AD. It remains unclear whether the disturbed composition of the microbiome and the domination of *Staphylococcus* in PN lesions may be the trigger factor of pruritus and lead to the development of cutaneous changes or is a secondary phenomenon. Our preliminary results support the theory that the composition of the skin microbiome in PN is altered and justify further research on the role of the microbiome in this debilitating condition.

## 1. Introduction

Prurigo nodularis (PN) is a chronic, debilitating condition characterized by the presence of multiple symmetrically distributed nodular lesions accompanied by intense pruritus and scratching behavior [1]. The nodules tend to develop on the extensor surfaces of the lower and upper extremities, and the trunk [2]. The disease most frequently affects older individuals, and is more common in females than in males [3]. The presence of the lesions and the accompanying itch exert significant burden on the quality of life [2]. PN has been associated with increased rates of depression and anxiety, as well as with numerous systemic comorbidities, including celiac disease, Hashimoto thyroiditis, inflammatory bowel diseases, type 1 and type 2 diabetes mellitus, chronic kidney disease, non-Hodgkin and Hodgkin lymphoma and primary cutaneous lymphoma [3,4,5,6,7]. In addition, about 65–80% of PN patients have a positive atopic background [8].

The exact etiology of PN remains elusive, which prevents the use of targeted treatment. Histopathology of the lesional skin typically shows a decreased density of intraepidermal nerve fibers, hyperplastic dermal nerve fibers and abundant inflammatory infiltrate composed of T lymphocytes, mast cells and eosinophils [9,10]. Complex interactions between the immune cells and neuronal circuity are suggested to play a major role in the pathogenesis of this debilitating condition [11]. PN has also been linked to several infectious factors, most commonly to HIV infection [12,13], and several reports confirm the improvement of PN in HIV-infected patients with antiretroviral therapy [14,15]. The relationship of PN with hepatitis C virus (HCV) infection has been less studied [16]. Although the direct causal link remains unknown, it has been hypothesized that immune dysregulation due to viral infection may be implicated in the etiopathogenesis of PN [3,17].

Data on the direct presence of infectious agents in the lesions of PN are scarce [18,19]. So far, the bacteria in the lesional skin have been identified using classical isolation techniques. In the study by Sharma et al. [20], *Staphyloccoccus aureus* was identified in skin swabs from 27 patients (100%) with PN. However, it should be taken into consideration that all patients in this group had a positive history of atopy and increased level of IgE. Mattila et al. [18] found acid fast bacilli in 28% of PN lesions by Ziehl–Neelsen staining of tissue sections and identified nontuberculous mycobacteria in 14% of nodules using classical culture techniques. The authors suggested that atypical mycobacteria may be implicated in the pathogenesis of PN. Nevertheless, these initial findings have not been objectively verified by other researchers to date.

The microbiome is defined as a pool of microorganisms in a given niche [21,22]. Complex interactions between the skin microbiota both shape the resident microbial community and prevent colonization of pathogenic bacteria [23,24]. Culture-dependent approaches to bacteria identification are often biased and do not reflect the true composition of microbiota, as fast-growing microorganisms may inhibit the growth of slow-growing microorganisms, and some species may not even be possible to breed under laboratory conditions [25,26]. The availability of next generation sequencing (NGS)-based techniques using 16S ribosomal RNA (rRNA) analysis has revolutionized the approach to analyze the role of microorganisms in numerous conditions, including AD, acne and rosacea [27,28].

We hypothesized that complex interactions between the microorganisms could modulate the perception of pruritus and contribute to the pathogenesis of PN. Therefore, the aim of this study was to evaluate the bacterial microbiome in PN lesions by targeting the region V3–V4 of 16S rRNA. In addition to patients with PN and healthy volunteers (HV), we also included patients with atopic dermatitis (AD), in whom the skin microbiome is disturbed and dominated by *S. aureus*.

## 2. Results

### 2.1. Participant Characteristic

The study involved 34 patients with prurigo nodularis, 15 patients with atopic dermatitis and 38 people without any dermatological disease. The microbiome genetic profile was extracted from 24 (70.6%) patients with PN (10 men and 14 women; mean age 60.5 ± 13.5 years); 14 (93.3%) patients with AD (3 men and 11 women; mean age 34.3 ± 16.9 years) and 9 (23.7%) HV (5 men and 4 women; mean age 56.9 ± 23.0 years). Detailed patient characteristics are presented in Table 1. Graphic presentation of bacteria found in all patients is demonstrated in Figure 1.

### 2.2. Alpha-Diversity

The Shannon Index showed no statistically significant difference between the three groups (*p* = 0.15). In paired analyses, there was no statistically significant difference between the PN and HV (*p* = 0.11), as well as between the AD and the control group (*p* = 0.06). The Shannon Index did not show a statistically significant difference between the PN and AD groups (*p* = 0.61).

On a global level, the Simpson Index showed no statistically significant difference between the three groups (*p* = 0.08). However, the paired analysis showed a statistically significant difference between the PN and HV (*p* < 0.05), as well as between the AD and the control group (*p* = 0.03). The alpha-diversity of the PN and AD microbiomes, assessed using Simpson Index, was highly similar and no statistically significant difference was observed between these two groups (*p* = 0.95).

### 2.3. Beta-Diversity

The Bray-Curtis distance showed a statistically significant difference between patients with PN, AD and HV, both on a global level (*p* = 0.001) and between the paired groups. The difference was statistically significant between the PN microbiome and HV (*p* = 0.001), as well as between the AD microbiome and HV (*p* = 0.001), and between the PN and AD groups (*p* = 0.018).

The results of the analysis using the Jaccarde distance were similar. The Jaccarde distance showed statistically significant differences between the three groups on a global level (*p* = 0.001) and between the paired groups. Further analysis showed statistically significant differences between the PN group and HV (*p* = 0.044), the AD microbiome and control group (*p* = 0.01), as well as between the PN and AD group (*p* = 0.02).

### 2.4. Relative Abundance

The ANCOM analysis was performed in order to select taxa differentiating individual groups. Twenty-four bacterial taxa were identified at the phylum level, 40 taxa at the class level, 120 taxa at the order level, 229 taxa at the family level and 565 taxa at the genus level. The ANCOM analysis showed that samples from PN and AD groups contained significantly more *Staphylococcus* that the samples from HV. This difference was maintained across all taxonomic levels. Interestingly, no taxa differentiating the PN microbiome from the AD microbiome were identified.

At the phylum level, *Firmicutes* predominated in the PN and AD microbiomes. On the other hand, *Proteobacteria* followed by *Actinobacteriota* were predominant in the microbiome of HV. At the class level, there was a predominance of *Bacilli* over *Gammaproteobacteria* in both PN and AD groups. The opposite was noted in HV—*Gammaproteobacteria* prevailed over *Bacilli*. At the genus level, *Staphylococcus* had a significantly greater share in the microbiome composition in PN and AD patients than in HV: *Staphylococcus* was identified as the most abundant in the PN microbiome (mean 81.57%) (Figure 1 and Figure 2). The other genus showed a lower abundance: *Pseudomonas* (5.77%), *Escherichia-Shigella* (4.35%), *Hymenobacter* (3.38%), *Microbacterium* (1.55%) and *Streptococcus* (1.32%). The cutaneous microbiome of AD patients was dominated by *Staphylococcus* (93.58%), followed by *Hymenobacter* (4.37%) and *Streptococcus* (0.91%) (Figure 3). The microbiome of HV was more diverse, with the predominance of *Pseudomonas* (50.72%) over *Staphylococcus* (16.43%), *Arthrobacter* (10.86%), *Finegoldia* (2.92%), *Nevskia* (2.16%), *Brochothrix* (2.14%), *Streptococcus* (1.93%), *Anaerococcus* (1.55%) and *Chloroplast* (1.41%) (Figure 4).

## 3. Discussion

Despite some evidence that infectious factors, both systemic and local, may contribute to the development of PN, the exact role of the microbiome in the pathogenesis of this entity has not been explored in detail so far. In the current study, we aimed to analyze and compare the diversity and composition of microorganisms from PN lesions to that of AD and healthy skin. We decided to include patients with AD in the analysis because it is well known that the skin microbiome in this entity is altered and that most patients are colonized with *S. aureus*. In this regard, the results of our research are consistent with data from the literature.

The current study highlights several important points. Firstly, the skin microbiome in PN lesions is definitely altered, both in terms of its diversity and composition, when compared to healthy skin. The relative abundance of *Staphylococcus* in patients with PN compared to controls is noteworthy. Secondly, the PN microbiome alterations show some similarities to those observed in AD. In both entities, the intra- and inter-sample diversity was decreased compared to healthy skin. In addition, *Staphylococcus* dominance was demonstrated in both diseases, which in the case of AD, is consistent with the results of previous studies [29,30,31]. We also know that the predominant bacterial species in AD is *Staphylococcus aureus*, while in PN, it is not entirely clear which *Staphylococcus* species actually dominates the skin lesions. However, our results may indicate that at some point, the skin microbiome in PN patients becomes similar to that found in AD leading to or accelerating type 2 inflammation. This may explain why dupilumab, a drug blocking IL4/IL13 initially designed for AD treatment, has also been revealed to effectively control PN symptoms [32].

Staphylococci are common colonizers of the skin in humans and the most important causes of bacterial skin infections. At present, the genus *Staphylococcus* comprises an estimated 50 different species. They are Gram-positive bacteria, which can be divided into two main groups: coagulase-positive staphylococci, with the most important species being *Staphylococcus aureus* and coagulase-negative staphylococci, which comprise most species including *Staphylococcus epidermidis*. On healthy human skin, *S. epidermidis* is more prevalent than *S. aureus* where certain *S. epidermidis* strains produce lipases and esterases, creating unfavorable growth conditions for pathogenic microorganisms. In addition, these bacteria have been shown to produce a protease that destroys *S. aureus* biofilms.

Nevertheless, *S. aureus* is a common inhabitant of the human microbiome and coexists with *Actinobacteria*, *Firmicutes*, *Bacteroidetes*, *Proteobacteria*, etc. on the skin and in the nasal passages [33]. Its colonization and ability to secrete a wide array of toxins and superantigens have been linked to the exacerbation of AD [34,35]. In particular, α- and δ-hemolysin and phenol-soluble modulins (PSMs) may lead to skin barrier disruption and promote AD-like inflammation [35]. However, not only is the presence of the microorganism itself important in the development of the disease, but above all, its metabolic and enzymatic activity. Hong et al. [36] observed a higher production of α-hemolysin in *S. aureus* collected from patients with a severe flare of AD compared to patients with mild or moderate disease. Moreover, the vast majority of *S. aureus* (91%) from patients with AD produced the α-hemolysin, while in healthy controls this percentage was only 33%. On the other hand, δ-hemolysin is a major virulence factor that is responsible for mast cell degranulation and, subsequently, the triggering of itch [37,38], while PSMs were demonstrated to assist in *S. aureus* biofilm formation and to induce production of pro-inflammatory cytokines in human keratinocytes [39,40]. PSMα3 was identified as the most potent of the PSM family, and it was demonstrated to upregulate a wide range of chemokines (e.g., CXCL1, CXCL2, CXCL3, CXCL5, CXCL8) and cytokines, including IL-1α, IL-1β, IL-6, IL-36γ and TNFα [40]. In summary, the virulence factors secreted during *S. aureus* colonization can trigger a significant immune response and contribute to the aggravation of cutaneous inflammation [38].

An impaired epidermal barrier allows numerous antigens, including pathogens, to penetrate through the stratum corneum to keratinocytes and Langerhans cells [41]. Exposure to *S. aureus* can initiate innate immune responses, increase the production of IL-33 and thymic stromal lymphopoietin (TSLP) and promote Th2-mediated inflammation [42].

The pathogenesis of PN has not been fully elucidated, yet. There is a general consensus that the development of cutaneous lesions results from a vicious circle of chronic itch and repeated scratching. IL-31 is considered to be the crucial cytokine in the pathogenesis of PN as its expression is associated with significant inflammation and pruritus, while anti-IL-31 antibodies have been demonstrated to significantly reduce the scratching activity [43]. It is worth noting that IL-31 can be upregulated by *S. aureus* superantigens [44].

However, the potential role of *Staphylococcus* in the pathogenesis of PN needs further research. It remains unclear whether the disturbed composition of the microbiome and domination of *Staphylococcus* in PN lesions may be the trigger factor of pruritus and lead to the development of cutaneous changes or is a secondary phenomenon. The origin of *Staphylococcus* (type of bacteria at species level, acquisition of new strains or relative overgrowth of preexisting strains) and its enzymatic activity in PN also remain unknown.

Our results support the theory that the composition of the skin microbiome in PN is altered and justifies further research on the role of the microbiome in this debilitating condition. The major limitation of this study is the small sample size and inability to identify bacteria at the species level. In addition, in this study, we tried to choose a method of collecting the material that was as minimally invasive as possible and at the same time the safest for the patient. Therefore, we abandoned the biopsy of skin lesions, which has been carried out in other research [18]; therefore, the analysis material of our study was only skin scrapings. Nevertheless, such a method does not guarantee obtaining a sufficient amount of material, especially from healthy skin. For this reason, the DNA profile of the bacterial microbiome failed to be extracted in most of the samples from healthy controls. Another limitation may be the amplification of the V3–V4 region of the bacterial 16S rRNA gene, which may not be the ideal region for microbiome evaluation of the skin and, possibly, the amplification of the V1–V3 could provide a higher richness of the microbiota composition, which needs to be tested in future studies [45].

Despite these limitations, to the best of our knowledge, this is the first metagenomic study on the skin microbiome in PN. Understanding the complex interactions between skin microbiota in PN may be an important step in developing target treatments with prebiotics and probiotics. The intention of future studies is to include a larger number of participants, taking into account various age groups and locations of the skin lesions.

## 4. Materials and Methods

### 4.1. Participant Recruitment

Consecutive adult patients with PN and AD who presented to the Department of Dermatology in Rzeszów (Poland) from March 2020 to January 2022 were recruited for the study. The diagnosis of PN was made according to the proposed criteria on the basis of the medical history and clinical presentation [46]. The clinical image of skin lesions in one of our patients suffering from PN is shown in Figure 5. If needed, the diagnosis was confirmed using histopathology. All participants with AD fulfilled the Hanifin–Rajka diagnostic criteria of AD. The exclusion criteria were as follows: concomitant dermatological or systemic conditions that might affect the microbiome composition, the use of systemic or topical antibiotics within the preceding 4 weeks, treatment of PN or AD with systemic or topical glicocorticosteroids, immunosuppressive agents or phototherapy within the preceding 12 weeks. Full medical history, including demographics, clinical course of the dermatological condition, concomitant diseases and medication, was taken. Each participant underwent a complete physical examination performed by a dermatologist. The intensity of the worst itch over the last 24 h in PN and AD groups was assessed using a numerical rating scale (WI-NRS) and the quality of life impairment—using the Dermatology Life Quality Index (DLQI). Controls were healthy volunteers (HV) who presented to the Outpatient Clinic for a routine nevi check-up.

### 4.2. Sample Collection

Samples for microbiome analysis were obtained from active skin lesions (nodules) on the extensor surfaces of the lower leg in patients with PN, active inflammatory lesions on lower legs in patients with AD and corresponding skin areas in HV. Before sampling, the designated skin areas were left untreated for at least 2 weeks. All participants were instructed to avoid washing this area for 24 h prior to material collection.

Samples for microbiome analysis were collected without prior cleaning of the skin surface. All procedures were performed using sterile gloves to avoid cross-contamination. A sterile ring of 2.5 cm in diameter was placed on the designated skin area (active nodule in PN, inflammatory patch in AD or corresponding skin site in HV). Approximately 0.5 mL of sterilized phosphate-buffered saline (PBS) was applied into the ring. The skin surface was slightly rubbed with a glass rod in a circular fashion (10 times to the left and 10 times to the right) to obtain PBS enriched in skin surface microbiota. Harvested PBS was stored at −70 °C for further analysis. At each time, a sample of PBS (0.5 mL) was collected in order to exclude contamination.

### 4.3. 16S rRNA Gene Sequencing and Analysis

DNA was extracted using Qiagen’s Pathogene Lysis Tubes (Hilden, Germany, Cat.No: 19092) and QIAamp UCP Pathogen Mini Kit (Hilden, Germany, Cat. No: 50214) and suspended in Buffer AVE according to the manufacturer’s protocol. The DNA concentration of the isolates was measured with a NanoDropTM2000 (ThermoScientific, Waltham, MA, USA). Total amounts of 7.7–20 ng/µL of DNA were obtained. The amplification of the V3-V4 region of the bacterial 16S rRNA gene was performed with the following primers:

16S-Amplicon-PCR-Forward-Primer: 5′TCGTCGGCAGCGTCAGATGTGTATAAGAGACAGCCTACGGGNGGCWGCAG

and 16S-Amplicon-PCR-Reverse-Primer: 5′GTCTCGTGGGCTCGGAGATGTGTATAAGAGACAGGACTACHVGGGTATCTAATCC.

Phanta Max Master Mix polymerase (Vazyme, Nankin, China) was used to perform the amplification. Components of the PCR reaction were as follows: 12.5 µL Polymerase, Primer 16S Amplicon PCR Forward 0.5 µL, Primer 16S Amplicon PCR Reverse 0.5 µL, DNA 11.5 µL. The reaction occurred on the C1000 TouchTermalCycler (Bio-Rad, Hercules, CA, USA) using the following program: 95 °C for 3 min followed by 25 cycles; 95 °C for 30 s, 55 °C for 30 s, 72 °C for 30 s, finalized with 72 °C for 5 min. The PCR products were verified on a 1.5% agarose gel. The samples that showed a band of 550 bp were further processed.

The PCR products were purified using AMPure XP magnetic beads (BeckmanCoulter, Brea, CA, USA) according to the protocol. The prepared matrix was re-amplified using the Nexter XT Index Kit indices (Illumina, San Diego, CA, USA) on the C1000 TouchTermalCycler thermocycler (Bio-Rad, Hercules, CA, USA) using the program: 95 °C for 3 min followed by 8 cycles: 95 °C for 30 s, 55 °C for 30 s, 72 °C for 30 s and 72 °C for 5 min. The PCR reaction consisted of 25 µL Phanta Max Master Mix polymerase (Vazyme, Nankin, China), 5 µL NexteraXTIndex Starter 1 (N7xx), 5 µL NexteraXTIndex Starter 2 (N5xx), 10 µL nuclease free water, 5 µL template.

DNA libraries were purified using AMPure XP magnetic beads (BeckmanCoulter, Brea, CA, USA) according to the protocol. DNA libraries were verified on a 1.5% agarose gel. Normalization was performed by measuring the light intensity of the bars using the Image Lab program (Version 6.1.0, © 2020, Bio-Rad Laboratories, Inc., Hercules, CA, USA). Sequencing was performed using the Illumina platform on the MiSeq instrument in the 2 × 250 paired readings mode. Adapters and low-quality sequences were removed from the raw readings using the trimgalore v 0.6.4 program with the parameters --length 100 --max_n 0 -q 25.

### 4.4. Data Analysis and Statistical Analysis

Further analyzes were carried out on the qiime2 v 2021.2 platform [47]. Raw data pretreatment to select ASV (Amplicon Sequence Variant) was performed with DADA2 with the parameters “--p-trim-left-f 15 --p-trunc-len-f 245 --p-trim-left-r 15 --p-trunc-len-r 235”.

Alpha- and beta-diversity metrics were generated with the core-metrics-phylogenetic plugin with a sampling depth of 8159. Alpha-diversity corresponds to the evenness and richness of microorganisms in a given environment and it is also referred to as intra-sample diversity. It is a measure of how diverse a single sample is, usually taking into account the number of different species observed. In the current study, alpha-diversity was evaluated using the Shannon index and Simpson index. Comparisons between groups were performed using the Qiime2 “diversity alpha-group-significance” plugin with the Kruskal–Wallis test for both “all groups” and “pairwise” tests. Beta-diversity is a reflection of the inter-sample diversity, i.e., the between-subject differences in microbiome composition over time and by location. Beta-diversity reflects differences in microbial composition between the samples. The beta-diversity was assessed using Bray–Curtis distance, Jaccard distance and Weighted UniFrac. For beta-diversity comparisons, the Qiime2 “diversity beta-group-significance” plugin with the PERMANOVA test was used.

Taxonomic classification was performed using the feature-classifier classify-consensus-vsearch plugin with --p-perc-identity 0.97 parameter based on the pre-formatted SILVA reference sequence and taxonomy files “Silva 138 99% OTUs full-length sequences” from Qiime2 data resources. The Analysis of Composition of Microbiomes (ANCOM) was used to identify features that differed in abundance between groups [48].

## Figures and Tables

**Figure 1 ijms-24-07675-f001:**
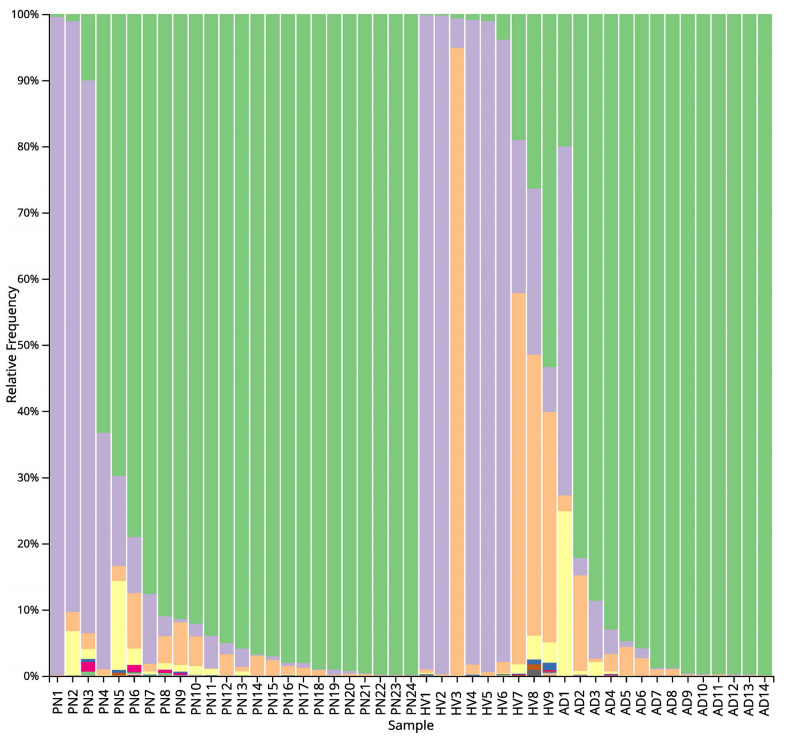
Stacked bar plots presenting the individual’s distribution of microorganisms in prurigo nodularis (PN), healthy volunteers (HV) and atopic dermatitis (AD) (**a**) at the phylum level (level 2), (**b**) at the class level (level 3) and (**c**) at the genus level (level 6).

**Figure 2 ijms-24-07675-f002:**
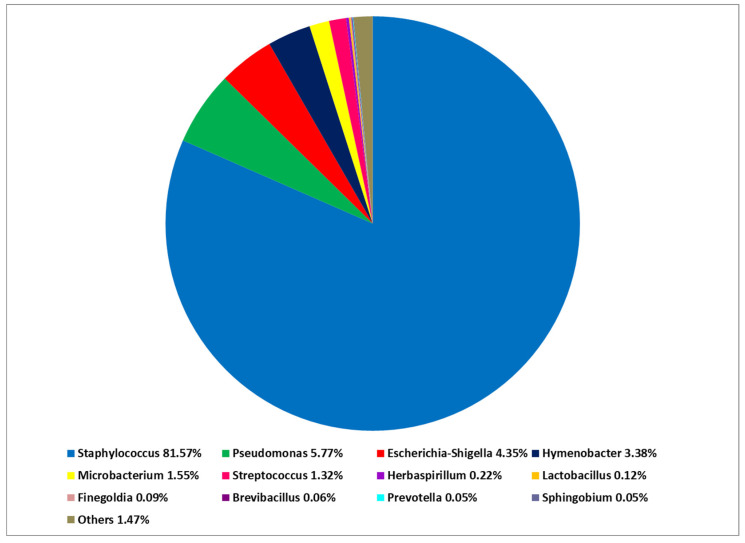
Total composition of cutaneous bacteria microbiome at the genus level in patients with prurigo nodularis (PN).

**Figure 3 ijms-24-07675-f003:**
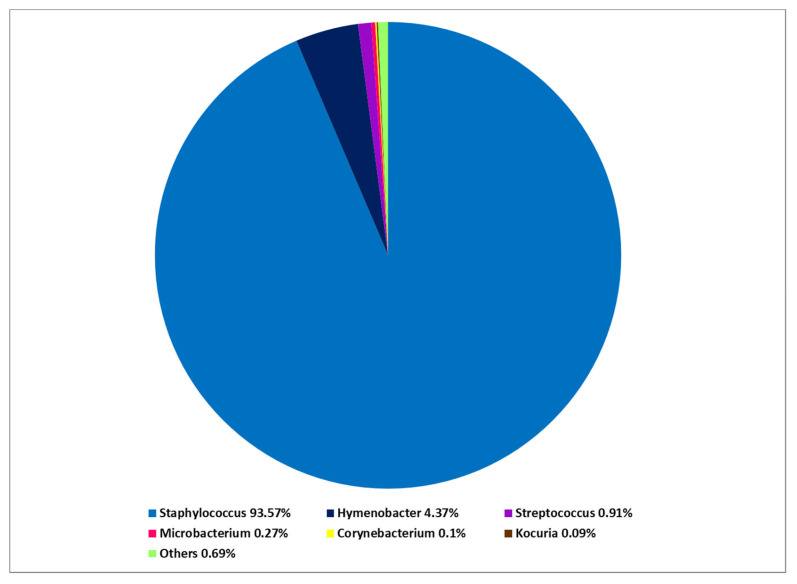
Total composition of the cutaneous bacteria microbiome at the genus level in patients with atopic dermatitis (AD).

**Figure 4 ijms-24-07675-f004:**
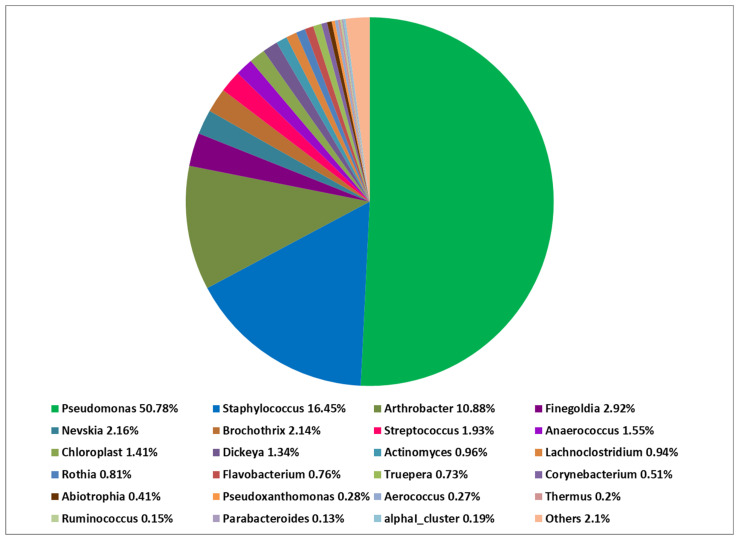
Total composition of the cutaneous bacteria microbiome at the genus level in healthy volunteers (HV).

**Figure 5 ijms-24-07675-f005:**
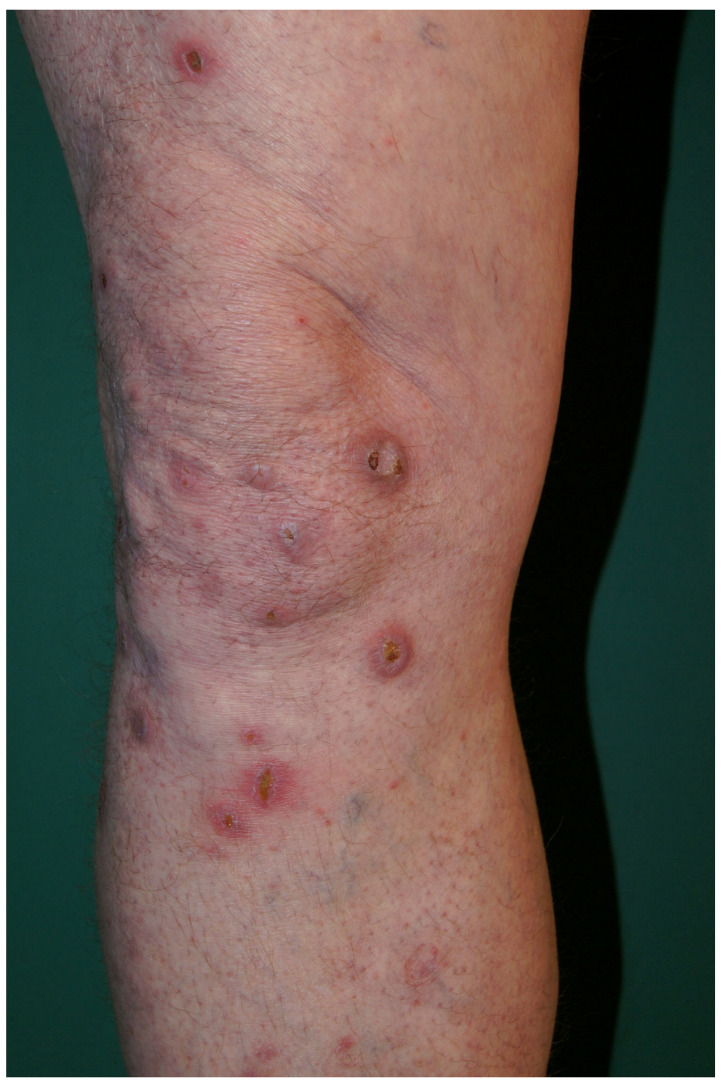
Clinical image of prurigo nodularis lesions.

**Table 1 ijms-24-07675-t001:** Clinical characteristics of study participants (PN—prurigo nodularis; AD—atopic dermatitis; HV—healthy volunteers; *n*—number of cases; SD—standard deviation).

	Patients with PN (*n* = 24)	Patients with AD (*n* = 14)	HV (*n* = 9)
Age, years			
Mean ± SD	60.5 ± 13.5	34.3 ± 16.9	56.9 ± 23.0
Range	29–78	18–64	24–88
Sex, *n* (%)			
Male	10 (41.7)	3 (21.4)	5 (55.6)
Female	14 (58.3)	11 (78.6)	4 (44.4)
Site of collection:			
lower leg, *n* (%)	24 (100)	14 (100)	9 (100)
Disease duration, years			
Mean ± SD	7.6 ± 6.8	19.6 ± 16.4	-
Range	0.75–25	1–64	-
Current exacerbation, months			
Mean ± SD	11.6 ± 14.4	7.2 ± 10.4	-
Range	0–48	0.25–36	-
History of atopic diseases, *n* (%)			
Mean ± SD	6.7 ± 2.9	8.0 ± 2.4	-
Range	0–10	2–10	-
DLQI, points			
Mean ± SD	8.8 ± 5.1	17.1 ± 4.3	-
Range	2–20	11–26	-
SCORAD	-	47.1–90.5 (63.0 ± 16.1)	-
Mean ± SD			
Range			
EASI	-	7.2–49.2 (31.4 ± 13.8)	-
Mean ± SD			
Range			
IGA	-	2–4 (3.1 ± 0.5)	-
Mean ± SD			
Range			
Dominant skin lesions		-	-
Nodule	19 (79.2%)		
Plaques	3 (12.5%)		
Erosion	1 (4.2%)		
Post-inflammatory discoloration or discoloration	1 (4.2%)		
Number of skin lesions		-	-
1–19	6 (25%)		
20–100	13 (54.2%)		
>100	5 (20.8%)		
Percentage of skin lesions with excoriations and crust		-	-
1–25	6 (25%)		
26–50	7 (29.2%)		
51–75	9 (37.5%)		
76–100	2 (8.3%)		
Activity of skin lesions		-	-
Almost unchanged	4 (16.7%)		
Mild	8 (33.3%)		
Moderate	11 (45.8%)		
Severe	1 (4.2%)		

## Data Availability

Anonymized source data generated during this study are available on the reasonable request from the corresponding authors.

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
