# Peer review of "Skin Microbiome in Prurigo Nodularis"

_ijms, 2023, doi:10.3390/ijms24087675_

Round 1
Reviewer 1 Report
dear author
very interesting information and article however main concern on this presentation : for skin the best for microbiome is V1-V3 as you know so please comment your choice. You mentioned sampling but knowing that body site are really driven the microbiome please mention the region sample and confirm that your analysis is nnot influence by body site. You analysed with diversity fonding that prurigo is clearly different than AD but you conclude that prurigo is close to AD please correct or explain. The level of species analysis is impossible using sequencing 16 S Sequencing using the Illumina platform on the MiSeq instrument; so you are unable to conclude on S aureus. please use WGMS if you wish. So really interesting topic but you should revised it to have it accurate .
Author Response
Thank you for reading our manuscript and providing us with valuable suggestions to improve our manuscript. The following changes have been made in the revised manuscript:
- Regarding region V3-V4, in fact, many researchers used the same region for skin microbiome analysis. For that reason we have decided to analyze it as well (Wongtada et al. Distinct skin microbiome modulation following different topical acne treatments in mild acne vulgaris patients: A randomized, investigator-blinded exploratory study. Exp Dermatol. 2023 Feb 26. doi: 10.1111/exd.14779; Guo HX et al. Microbe community composition differences of hand skin on similar lifestyle volunteers: a small-scale study. J Appl Microbiol. 2023 Feb 16;134(2):lxac068; Tian Y et al. Alteration of Skin Microbiome in CKD Patients Is Associated With Pruritus and Renal Function. Front Cell Infect Microbiol. 2022 Jun 28;12:923581).
- We have provided the localization of sampling, i.e. lower legs, as this is the most common region of prurigo lesions.
- Initially, we had the idea, that the microbiome will be significantly different in PN compared with AD, but finally, we found significant similarities. We have added some additional sentences to the discussion.
- We have changed the data presentation to genus level.
We hope, that the revised version will be found suitable for publication.
Reviewer 2 Report
The paper submitted by Tutka and colleagues demonstrated the skin microbiome community among PN patients, AD patients, and healthy volunteers, using 16S illumina sequencing technology. The sample preparation is very neat and scientific.
1. The Figure legends need to be improved. For the x-axis in Fig. 2 and the y-axis in Fig. 3, are confused to understand. If they are the number for PN patients or other group, I think authors need to point it out or show it on the figures to let audiences understand it.
2. Based on a lot of publications, the species level analysis using the 16S variable regions with short-read sequencing (e.g. 16S illumina sequencing) is not reliable. (Johnson, J.S., Spakowicz, D.J., Hong, BY. et al. Evaluation of 16S rRNA gene sequencing for species and strain-level microbiome analysis. Nat Commun 10, 5029 (2019). https://doi.org/10.1038/s41467-019-13036-1).
I highly recommended the authors reconsider doing the species-level analysis. If you need to analyze your data at the species level, then you probably want to do full-length 16S rRNA sequencing to confirm that S. aureus is in the microbiome. There are different methods to confirm the relative abundance in the microbial community using full-length 16S rRNA sequencing, the authors can choose their own method to confirm it.
Author Response
Thank you for reading our manuscript and providing us with valuable suggestions to improve our manuscript. The following changes have been made in the revised manuscript:
- Figures were redesigned and x-axis and y-axis were modified.
- We have changed the data presentation to genus level.
Reviewer 3 Report
I have read the paper with great interest but have some concerns which need to be addressed.
1. The skin microbiome differs greatly between people and is highly affected by age, sex, sampling location etc. The authors have compared the skin microbiota between PN, AD and healthy control who vary in age and sex. Also, the location of the skin scraping has not even been mentioned.
2. The authors collected samples by performing skin scraping. Why was this method chosen over skin swabs? With skin scraping, the skin is more likely to bleed and the sample can be contaminated with blood which makes the analysis more difficult. The low gene extraction rate questions the validity of the sampling method.
3. With the V3-4 sequencing, species level classification is not valid. Also, the tables and figure legends are hard to read.
I feel that a larger number of samples and age, sex, location matching is needed to correctly draw a conclusion.
Author Response
We would like to thank the reviewer for his efforts to read and comment on our article. We are also thankful for any suggestions to improve our manuscript. We have revised our report strictly according to the reviewer"s comments.
- Indeed, the microbiome may vary greatly depending on the location of the skin sampling. Having this in mind, in fact, we have collected all our samples from the lower legs as this is the most common location of skin lesions in PN patients (our target group).
- We would like to apologize for being not enough precise in our original report. In fact, it was rather a delicate rubbing and not scraping while performing the skin sampling. In the revised manuscript this mistake was corrected. We also agree with the reviewer, that the rate of DNA isolation, particularly from healthy volunteers was quite low. One explanation may be the method of skin sampling which had to be absolutely noninvasive to avoid blood contamination. Secondly, healthy skin was absolutely normal which may in part explain, why in many patients we were unable to collect enough valid bacterial DNA (intact epidermal barrier, no scaling, etc.). However, we believe, that at least in terms of AD and PN we collected a representative group of patients. Furthermore, the validity of the study could be confirmed by the AD microbiome which is similar to that found by other researchers.
- Figures were modified to make them more readable. The species level was removed from the manuscript. We have provided the analysis of the genus level instead (level 6).
- We agree that a larger patient number would make our study more credible, however, we feel that our data are anyway interesting, as we demonstrate for the first time, that microbiome in PN is rather similar to AD - a phenomenon, that was not reported previously. We do believe, that our findings may encourage other researchers to analyse skin microbiome in further PN patients in different regions and ethnic groups.
Round 2
Reviewer 1 Report
the corrections are fine. please add in the discussion that choosen the V3V4 is not the best method for skin microbiome evaluation and V1 V3 would have been better.
Author Response
Dear Reviewer,
Thank you for your suggestion. We have added this statement in the discussion (marked in red).
Reviewer 2 Report
I like the modification from the authors. I think it is now hit the publish requirements.
Author Response
We are grateful to the reviewer for the supportive thoughts.
Reviewer 3 Report
Thank you for revising the paper. I am satisfied with the revision.
Author Response
Dear Reviewer,
Thank you for your support.